# Clustering Billions of Reads for DNA Data Storage

**Cyrus Rashtchian**[a,b]   **Konstantin Makarychev**[a,c]   **Miklós Rácz**[a,d]   **Siena Dumas Ang**[a]
**Djordje Jevdjic**[a]   **Sergey Yekhanin**[a]   **Luis Ceze**[a,b]   **Karin Strauss**[a]
[a]Microsoft Research,  [b]CSE at University of Washington,
[c]EECS at Northwestern University,  [d]ORFE at Princeton University

## Abstract

Storing data in synthetic DNA offers the possibility of improving information density and durability by several orders of magnitude compared to current storage technologies. However, DNA data storage requires a computationally intensive process to retrieve the data. In particular, a crucial step in the data retrieval pipeline involves clustering billions of strings with respect to edit distance. Datasets in this domain have many notable properties, such as containing a very large number of small clusters that are well-separated in the edit distance metric space. In this regime, existing algorithms are unsuitable because of either their long running time or low accuracy. To address this issue, we present a novel distributed algorithm for approximately computing the underlying clusters. Our algorithm converges efficiently on any dataset that satisfies certain separability properties, such as those coming from DNA data storage systems. We also prove that, under these assumptions, our algorithm is robust to outliers and high levels of noise. We provide empirical justification of the accuracy, scalability, and convergence of our algorithm on real and synthetic data. Compared to the state-of-the-art algorithm for clustering DNA sequences, our algorithm simultaneously achieves higher accuracy and a 1000x speedup on three real datasets.

## 1   Introduction

Existing storage technologies cannot keep up with the modern data explosion. Thus, researchers have turned to fundamentally different physical media for alternatives. Synthetic DNA has emerged as a promising option, with theoretical information density of multiple orders of magnitude more than magnetic tapes [12, 24, 26, 52]. However, significant biochemical and computational improvements are necessary to scale DNA storage systems to read/write exabytes of data within hours or even days.

Encoding a file in DNA requires several preprocessing steps, such as randomizing it using a pseudo-random sequence, partitioning it into hundred-character substrings, adding address and error correction information to these substrings, and finally encoding everything to the $\{A, C, G, T\}$ alphabet. The resulting collection of short strings is synthesized into DNA and stored until needed.

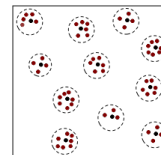

To retrieve the data, the DNA is accessed using next-generation sequencing, which results in several noisy copies, called *reads*, of each originally synthesized short string, called a *reference*. With current technologies, these references and reads contain hundreds of characters, and in the near future, they will likely contain thousands [52]. After sequencing, the goal is to recover the unknown references from the observed reads. The first step, which is the focus of this paper, is to cluster the reads into groups, each of which is the set of noisy copies of a single reference.

Figure 1: DNA storage datasets have many small clusters that are well-separated in edit distance.

The output of clustering is fed into a consensus-finding algorithm, which predicts the most likely reference to have produced each cluster of reads. As Figure 1 shows,

datasets typically contain only a handful of reads for each reference, and each of these reads differs from the reference by insertions, deletions, and/or substitutions. The challenge of clustering is to achieve high precision and recall of many small underlying clusters, in the presence of such errors.

Datasets arising from DNA storage have two striking properties. First, the number of clusters grows linearly with the input size. Each cluster typically consists of five to fifteen noisy copies of the same reference. Second, the clusters are separated in edit distance, by design (via randomization). We investigate approximate clustering algorithms for large collections of reads with these properties.

Suitable algorithms must satisfy several criteria. First, they must be distributed, to handle the billions of reads coming from modern sequencing machines. Second, their running time must scale favorably with the number of clusters. In DNA storage datasets, the size of the clusters is fixed and determined by the number of reads needed to recover the data. Thus, the number of clusters $k$ grows linearly with the input size $n$ (i.e., $k = \Omega(n)$). Any methods requiring $\Omega(k \cdot n) = \Omega(n^2)$ time or communication would be too slow for billion-scale datasets. Finally, algorithms must be robust to noise and outliers, and they must find clusters with relatively large diameters (e.g., linear in the dimensionality).

These criteria rule out many clustering methods. Algorithms for $k$-medians and related objectives are unsuitable because they have running time or communication scaling with $k \cdot n$ [19, 29, 33, 42]. Graph clustering methods, such as correlation clustering [4, 9, 18, 47], require a similarity graph.[1] Constructing this graph is costly, and it is essentially equivalent to our clustering problem, since in DNA storage datasets, the similarity graph has connected components that are precisely the clusters of noisy reads. Linkage-based methods are inherently sequential, and iteratively merging the closest pair of clusters takes quadratic time. Agglomerative methods that are robust to outliers do not extend to versions that are distributed and efficient in terms of time, space, and communication [2, 8].

Turning to approximation algorithms, tools such as metric embeddings [43] and locality sensitive hashing (LSH) [31] trade a small loss in accuracy for a large reduction in running time. However, such tools are not well understood for edit distance [16, 17, 30, 38, 46], even though many methods have been proposed [15, 27, 39, 48, 54]. In particular, no published system has demonstrated the potential to handle billions of reads, and no efficient algorithms have experimental or theoretical results supporting that they would achieve high enough accuracy on DNA storage datasets. This is in stark contrast to set similarity and Hamming distance, which have many positive results [13, 36, 40, 49, 55].

Given the challenges associated with existing solutions, we ask two questions: (1) Can we design a distributed algorithm that converges in sub-quadratic time for DNA storage datasets? (2) Is it possible to adapt techniques from metric embeddings and LSH to cluster billions of strings in under an hour?

**Our Contributions** We present a distributed algorithm that clusters billions of reads arising from DNA storage systems. Our agglomerative algorithm utilizes a series of filters to avoid unnecessary distance computations. At a high level, our algorithm iteratively merges clusters based on random representatives. Using a hashing scheme for edit distance, we only compare a small subset of representatives. We also use a light-weight check based on a binary embedding to further filter pairs. If a pair of representatives passes these two tests, edit distance determines whether the clusters are merged. Theoretically and experimentally, our algorithm satisfies four desirable properties.

**Scalability:** Our algorithm scales well in time and space, in shared-memory and shared-nothing environments. For $n$ input reads, each of $P$ processors needs to hold only $O(n/P)$ reads in memory.

**Accuracy:** We measure accuracy as the fraction of clusters with a majority of found members and no false positives. Theoretically, we show that the separation of the underlying clusters implies our algorithm converges quickly to a correct clustering. Experimentally, a small number of communication rounds achieve 98% accuracy on multiple real datasets, which suffices to retrieve the stored data.

**Robustness:** For separated clusters, our algorithm is optimally robust to adversarial outliers.

**Performance:** Our algorithm outperforms the state-of-the-art clustering method for sequencing data, Starcode [57], achieving higher accuracy with a 1000x speedup. Our algorithm quickly recovers clusters with large diameter (e.g., 25), whereas known string similarity search methods perform poorly with distance threshold larger than four [35, 53]. Our algorithm is simple to implement in any distributed framework, and it clusters 5B reads with 99% accuracy in 46 minutes on 24 processors.

## 1.1 Outline

The rest of the paper is organized as follows. We begin, in Section 2, by defining the problem statement, including clustering accuracy and our data model. Then, in Section 3, we describe our algorithm, hash function, and binary signatures. In Section 4, we provide an overview of the theoretical analysis, with most details in the appendix. In Section 5, we empirically evaluate our algorithm. We discuss related work in Section 6 and conclude in Section 7.

## 2 DNA Data Storage Model and Problem Statement

For an alphabet $\Sigma$, the *edit distance* between two strings $x, y \in \Sigma^*$ is denoted $d_E(x, y)$ and equals the minimum number of insertions, deletions, or substitutions needed to transform $x$ to $y$. It is well known that $d_E$ defines a metric. We fix $\Sigma = \{\mathsf{A}, \mathsf{C}, \mathsf{G}, \mathsf{T}\}$, representing the four DNA nucleotides. We define the distance between two nonempty sets $C_1, C_2 \subseteq \Sigma^*$ as $d_E(C_1, C_2) = \min_{x \in C_1, y \in C_2} d_E(x, y)$. A *clustering* $\mathbf{C}$ of a finite set $S \subseteq \Sigma^*$ is any partition of $S$ into nonempty subsets.

We work with the following definition of accuracy, motivated by DNA storage data retrieval.

**Definition 2.1** (Accuracy)**.** Let $\mathbf{C}, \widetilde{\mathbf{C}}$ be clusterings. For $1/2 < \gamma \leqslant 1$ the *accuracy* of $\widetilde{\mathbf{C}}$ with respect to $\mathbf{C}$ is

$$\mathcal{A}_\gamma(\mathbf{C}, \widetilde{\mathbf{C}}) = \max_\pi \frac{1}{|\mathbf{C}|} \sum_{i=1}^{|\mathbf{C}|} \mathbf{1}\{\widetilde{C}_{\pi(i)} \subseteq C_i \ \text{ and } \ |\widetilde{C}_{\pi(i)} \cap C_i| \geqslant \gamma |C_i|\},$$

where the max is over all injective maps $\pi : \{1, 2, \dots, |\widetilde{\mathbf{C}}|\} \to \{1, 2, \dots, \max(|\mathbf{C}|, |\widetilde{\mathbf{C}}|)\}$.

We think of $\mathbf{C}$ as the underlying clustering and $\widetilde{\mathbf{C}}$ as the output of an algorithm. The accuracy $\mathcal{A}_\gamma(\mathbf{C}, \widetilde{\mathbf{C}})$ measures the number of clusters in $\widetilde{\mathbf{C}}$ that overlap with some cluster in $\mathbf{C}$ in at least a $\gamma$-fraction of elements while containing no false positives.[2] This is a stricter notion than the standard classification error [8, 44]. Notice that our accuracy definition does not require that the clusterings be of the same set. We will use this to compare clusterings of $S$ and $S \cup \mathsf{O}$ for a set of outliers $\mathsf{O} \subseteq \Sigma^*$.

For DNA storage datasets, the underlying clusters have a natural interpretation. During data retrieval, several molecular copies of each original DNA strand (reference) are sent to a DNA sequencer. The output of sequencing is a small number of noisy reads of each reference. Thus, the reads that correspond to the same reference form a cluster. This interpretation justifies the need for high accuracy: each underlying cluster represents one stored unit of information.

**Data Model**    To aid in the design and analysis of clustering algorithms for DNA data storage, we introduce the following natural generative model. First, pick many random centers (representing original references), then perturb each center by insertions, deletions, and substitutions to acquire the elements of the cluster (representing the noisy reads). We model the original references as random strings because during the encoding process, the original file has been randomized using a fixed pseudo-random sequence [45]. We make this model precise, starting with the perturbation.

**Definition 2.2** ($p$-noisy copy)**.** For $p \in [0, 1]$ and $z \in \Sigma^*$, define a $p$-noisy copy of $z$ by the following process. For each character in $z$, independently, do one of the following four operations: (i) keep the character unchanged with probability $(1 - p)$, (ii) delete it with probability $p/3$, (iii) with probability $p/3$, replace it with a character chosen uniformly at random from $\Sigma$, or (iv) with probability $p/3$, keep the character and insert an additional one after it, chosen uniformly at random from $\Sigma$.

We remark that our model and analysis can be generalized to incorporate separate deletion, insertion, and substitution probabilities $p = p_D + p_I + p_S$, but we use balanced probabilities $p/3$ to simplify the exposition. Now, we define a noisy cluster. For simplicity, we assume uniform cluster sizes.

**Definition 2.3** (Noisy cluster of size $s$)**.** We define the distribution $\mathcal{D}_{s,p,m}$ with cluster size $s$, noise rate $p \in [0, 1]$, and dimension $m$. Sample a cluster $C \sim \mathcal{D}_{s,p,m}$ as follows: pick a center $z \in \Sigma^m$ uniformly at random; then, each of the $s$ elements of $C$ will be an independent $p$-noisy copy of $z$.

With our definition of accuracy and our data model in hand, we define the main clustering problem.

**Problem Statement**   Fix $p, m, s, n$. Let $\mathbf{C} = \{C_1, \ldots, C_k\}$ be a set of $k = n/s$ independent clusters $C_i \sim \mathcal{D}_{s,p,m}$. Given an accuracy parameter $\gamma \in (1/2, 1]$ and an error tolerance $\varepsilon \in [0, 1]$, on input set $S = \cup_{i=1}^{k} C_i$, the goal is to quickly find a clustering $\widetilde{\mathbf{C}}$ of $S$ with $\mathcal{A}_\gamma(\mathbf{C}, \widetilde{\mathbf{C}}) \geqslant 1 - \varepsilon$.

## 3   Approximately Clustering DNA Storage Datasets

Our distributed clustering method iteratively merges clusters with similar representatives, alternating between local clustering and global reshuffling. At the core of our algorithm is a hash family that determines (i) which pairs of representatives to compare, and (ii) how to repartition the data among the processors. On top of this simple framework, we use a cheap pre-check, based on the Hamming distance between binary signatures, to avoid many edit distance comparisons. Our algorithm achieves high accuracy by leveraging the fact that DNA storage datasets contain clusters that are well-separated in edit distance. In this section, we will define separated clusterings, explain the hash function and the binary signature, and describe the overall algorithm.

### 3.1   Separated Clusters

The most important consequence of our data model $\mathcal{D}_{s,p,m}$ is that the clusters will be well-separated in the edit distance metric space. Moreover, this reflects the actual separation of clusters in real datasets. To make this precise, we introduce the following definition.

**Definition 3.1.**   A clustering $\{C_1, \ldots, C_k\}$ is $(r_1, r_2)$-*separated* if $C_i$ has diameter[3] at most $r_1$ for every $i \in \{1, 2, \ldots, k\}$, while any two different clusters $C_i$ and $C_j$ satisfy $d_E(C_i, C_j) > r_2$.

DNA storage datasets will be separated with $r_2 \gg r_1$. Thus, recovering the clusters corresponds to finding pairs of strings with distance at most $r_1$. Whenever $r_2 \geqslant 2 \cdot r_1$, our algorithm will be robust to outliers. In Section 4, we provide more details about separability under our DNA storage data model. We remark that our clustering separability definition differs slightly from known notions [2, 3, 8] in that we explicitly bound both the diameter of clusters and distance between clusters.

### 3.2   Hashing for Edit Distance

Algorithms for string similarity search revolve around the simple fact that when two strings $x, y \in \Sigma^m$ have edit distance at most $r$, then they share a substring of length at least $m/(r + 1)$. However, insertions and deletions imply that the matching substrings may appear in different locations. Exact algorithms build inverted indices to find matching substrings, and many optimizations have been proposed to exactly find all close pairs [34, 51, 57]. Since we need only an approximate solution, we design a hash family based on finding matching substrings quickly, without being exhaustive. Informally, for parameters $w, \ell$, our hash picks a random "anchor" $a$ of length $w$, and the hash value for $x$ is the substring of length $w + \ell$ starting at the first occurrence of $a$ in $x$.

We formally define the family of hash functions $\mathcal{H}_{w,\ell} = \{h_{\pi,\ell} : \Sigma^* \to \Sigma^{w+\ell}\}$ parametrized by $w, \ell$, where $\pi$ is a permutation of $\Sigma^w$. For $x = x_1 x_2 \cdots x_m$, the value of $h_{\pi,\ell}(x)$ is defined as follows. Find the earliest, with respect to $\pi$, occurring $w$-gram $a$ in $x$, and let $i$ be the index of the first occurrence of $a$ in $x$. Then, $h_{\pi,\ell}(x) = x_i \cdots x_{m'}$ where $m' = \min(m, i + w + \ell)$. To sample $h_{\pi,\ell}$ from $\mathcal{H}_{w,\ell}$, simply pick a uniformly random permutation $\pi : \Sigma^w \to \Sigma^w$.

Note that $\mathcal{H}_{w,\ell}$ resembles MinHash [13, 14] with the natural mapping from strings to sets of substrings of length $w + \ell$. Our hash family has the benefit of finding long substrings (such as $w + \ell = 16$), while only having the overhead of finding anchors of length $w$. This reduces computation time, while still leading to effective hashes. We now describe the signatures.

### 3.3   Binary Signature Distance

The $q$-gram distance is an approximation for edit distance [50]. By now, it is a standard tool in bioinformatics and string similarity search [27, 28, 48, 54]. A $q$-gram is simply a substring of length $q$, and the $q$-gram distance measures the number of different $q$-grams between two strings. For a string

**Algorithm 1** Clustering DNA Strands

---

1: **function** CLUSTER($S$, $r$, $q$, $w$, $\ell$, $\theta_{low}$, $\theta_{high}$, comm_steps, local_steps)
2:      $\widetilde{\mathbf{C}} = S$.
3:      **For** $i = 1, 2, \ldots,$ comm_steps:
4:          Sample $h_{\pi,\ell} \sim \mathcal{H}_{w,\ell}$ and hash-partition clusters, applying $h_{\pi,\ell}$ to representatives.
5:          **For** $j = 1, 2, \ldots,$ local_steps:
6:              Sample $h_{\pi,\ell} \sim \mathcal{H}_{w,\ell}$.
7:              **For** $C \in \widetilde{\mathbf{C}}$, sample a representative $x_C \sim C$, and then compute the hash $h_{\pi,\ell}(x_C)$.
8:              **For** each pair $x, y$ with $h_{\pi,\ell}(x) = h_{\pi,\ell}(y)$:
9:                  **If** $(d_H(\sigma(x), \sigma(y)) \leqslant \theta_{low})$ or $(d_H(\sigma(x), \sigma(y)) \leqslant \theta_{high}$ and $d_E(x, y) \leqslant r)$:
10:                     Update $\widetilde{\mathbf{C}} = (\widetilde{\mathbf{C}} \setminus \{C_x, C_y\}) \cup \{C_x \cup C_y\}$.
11:     **return** $\widetilde{\mathbf{C}}$.
12: **end function**

---

$x \in \Sigma^m$, let the binary signature $\sigma_q(x) \in \{0, 1\}^{4^q}$ be the indicator vector for the set $q$-grams in $x$. Then, the $q$-gram distance between $x$ and $y$ equals the Hamming distance $d_H(\sigma_q(x), \sigma_q(y))$.

The utility of the $q$-gram distance is that the Hamming distance $d_H(\sigma_q(x), \sigma_q(y))$ approximates the edit distance $d_E(x, y)$, yet it is much faster to check $d_H(\sigma_q(x), \sigma_q(y)) \leqslant \theta$ than to verify $d_E(x, y) \leqslant r$. The only drawback of the $q$-gram distance is that it may not faithfully preserve the separation of clusters, in the worst case. This implies that the $q$-gram distance by itself is not sufficient for clustering. Therefore, we use binary signatures as a coarse filtering step, but reserve edit distance for ambiguous merging decisions. We provide theoretical bounds on the $q$-gram distance in Section 4.1 and Appendix B. We now explain our algorithm.

### 3.4 Algorithm Description

We describe our distributed, agglomerative clustering algorithm (displayed in Algorithm 1). The algorithm ingests the input set $S \subset \Sigma^*$ in parallel, so each core begins with roughly the same number of reads. Signatures $\sigma_q(x)$ are pre-computed and stored for each $x \in S$. The clustering $\widetilde{\mathbf{C}}$ is initialized as singletons. It will be convenient to use the notation $x_C$ for an element $x \in C$, and the notation $C_x$ for the cluster that $x$ belongs to. We abuse notation and use $\widetilde{\mathbf{C}}$ to denote the current global clustering. The algorithm alternates between global communication and local computation.

**Communication**   One representative $x_C$ is sampled uniformly from each cluster $C_x$ in the current clustering $\widetilde{\mathbf{C}}$, in parallel. Then, using shared randomness among all cores, a hash function $h_{\pi,\ell}$ is sampled from $\mathcal{H}_{w,\ell}$. Using this same hash function for each core, a hash value is computed for each representative $x_C$ for cluster $C$ in the current clustering $\widetilde{\mathbf{C}}$. The communication round ends by redistributing the clusters randomly using these hash values. In particular, the value $h_{\pi,\ell}(x_c)$ determines which core receives $C$. The current clustering $\widetilde{\mathbf{C}}$ is thus repartitioned among cores.

**Local Computation**   The local computation proceeds independently on each core. One local round revolves around one hash function $h_{\pi,\ell} \sim \mathcal{H}_{w,\ell}$. Let $\widetilde{\mathbf{C}}_j$ be the set of clusters that have been distributed to the $j$th core. During each local clustering step, one uniform representative $x_C$ is sampled for each cluster $C \in \widetilde{\mathbf{C}}_j$. The representatives are bucketed based on $h_{\pi,\ell}(x_c)$. Now, the local clustering requires three parameters, $r, \theta_{low}, \theta_{high}$, set ahead of time, and known to all the cores. For each pair $y, z$ in a bucket, first the algorithm checks whether $d_H(\sigma_q(y), \sigma_q(z)) \leqslant \theta_{low}$. If so, the clusters $C_y$ and $C_z$ are merged. Otherwise, the algorithm checks if both $d_H(\sigma_q(y), \sigma_q(z)) \leqslant \theta_{high}$ and $d_E(x, y) \leqslant r$, and merges the clusters $C_y$ and $C_z$ if these two conditions hold. Immediately after a merge, $\widetilde{\mathbf{C}}_j$ is updated, and $C_x$ corresponds to the present cluster containing $x$. Note that distributing the clusters among cores during communication implies that no coordination is needed after merges. The local clustering repeats for $local\_steps$ rounds before moving to the next communication round.

**Termination**   After the local computation finishes, after the last of $comm\_steps$ communication rounds, the algorithm outputs the current clustering $\widetilde{\mathbf{C}} = \bigcup_j \widetilde{\mathbf{C}}_j$ and terminates.

# 4 Theoretical Algorithm Analysis

## 4.1 Cluster Separation and Binary Signatures

When storing data in DNA, the encoding process leads to clusters with nearly-random centers. Recall that we need the clusters to be far apart for our algorithm to perform well. Fortunately, random cluster centers will have edit distance $\Omega(m)$ with high-probability. Indeed, two independent random strings have expected edit distance $c_{\text{ind}} \cdot m$, for a constant $c_{\text{ind}} > 0$. Surprisingly, the exact value of $c_{\text{ind}}$ remains unknown. Simulations suggest that $c_{\text{ind}} \approx 0.51$, and it is known that $c_{\text{ind}} > 0.338$ [25].

When recovering the data, DNA storage systems receive clusters that consist of $p$-noisy copies of the centers. In particular, two reads inside of a cluster will have edit distance $O(pm)$, since they are $p$-noisy copies of the same center. Therefore, any two reads in different clusters will be far apart in edit distance whenever $p \ll c_{\text{ind}}$ is a small enough constant. We formalize these bounds and provide more details, such as high-probability results, in Appendix A.

Another feature of our algorithm is the use of binary signatures. To avoid incorrectly merging distinct clusters, we need the clusters to be separated according to $q$-gram distance. We show that random cluster centers will have $q$-gram distance $\Omega(m)$ when $q = 2\log_4 m$. Additionally, for any two reads $x, y$, we show that $d_H(\sigma_q(x), \sigma_q(y)) \leqslant 2q \cdot d_E(x, y)$, implying that if $x$ and $y$ are in the same cluster, then their $q$-gram distance will be at most $O(qpm)$. Therefore, whenever $p \ll 1/q \approx 1/\log m$, signatures will already separate clusters. For larger $p$, we use the pair of thresholds $\theta_{low} < \theta_{high}$ to mitigate false merges. We provide more details in Appendix B.

In Section 5, we mention an optimization for the binary signatures, based on blocking, which empirically improves the approximation quality, while reducing memory and computational overhead.

## 4.2 Convergence and Hash Analysis

The running time of our algorithm depends primarily on the number of iterations and the total number of comparisons performed. The two types of comparisons are edit distance computations, which take time $O(rm)$ to check distance at most $r$, and $q$-gram distance computations, which take time linear in the signature length. To avoid unnecessary comparisons, we partition cluster representatives using our hash function and only compare reads with the same hash value. Therefore, we bound the total number of comparisons by bounding the total number of hash collisions. In particular, we prove the following convergence theorem (details appear in Appendix C.

**Theorem 4.1** (Informal). *For sufficiently large $n$ and $m$ and small $p$, there exist parameters for our algorithm such that it outputs a clustering with accuracy $(1 - \varepsilon)$ and the expected number of comparisons is*

$$O\left(\max\left\{n^{1+O(p)}, \frac{n^2}{m^{\Omega(1/p)}}\right\} \cdot \left(1 + \frac{\log(s/\varepsilon)}{s}\right)\right).$$

Note that $n^{1+O(p)} \geqslant n^2/m^{\Omega(1/p)}$ in the expression above whenever the reads are long enough, that is, when $m \geqslant n^{cp}$ (where $c$ is some small constant). Thus, for a large range of $n, m, p$, and $\varepsilon$, our algorithm converges in time proportional to $n^{1+O(p)}$, which is sub-quadratic in $n$, the number of input reads. Since we expect the number of clusters $k$ to be $k = \Omega(n)$, our algorithm outperforms any methods that require time $\Omega(kn) = \Omega(n^2)$ in this regime.

The running time analysis of our algorithm revolves around estimating both the collision probability of our hash function and the overall convergence time to identify the underlying clusters. The main overhead comes from unnecessarily comparing reads that belong to different clusters. Indeed, for pairs of reads inside the same cluster, the total number of comparisons is $O(n)$, since after a comparison, the reads will merge into the same cluster. For reads in different clusters, we show that they collide with probability that is exponentially small in the hash length (since they are nearly-random strings). For the convergence analysis, we prove that reads in the same cluster will collide with significant probability, implying that after roughly

$$O\left(\max\left\{n^{O(p)}, \frac{n}{m^{\Omega(1/p)}}\right\} \cdot \left(1 + \frac{\log(s/\varepsilon)}{s}\right)\right)$$

iterations, the found clustering will be $(1 - \varepsilon)$ accurate.

In Section 5, we experimentally validate our algorithm's running time, convergence, and correctness properties on real and synthetic data.

### 4.3  Outlier Robustness

Our final theoretical result involves bounding the number of incorrect merges caused by potential outliers in the dataset. In real datasets, we expect some number of highly-noisy reads, due to experimental error. Fortunately, such outliers lead to only a minor loss in accuracy for our algorithm, when the clusters are separated. We prove the following theorem in Appendix D.

**Theorem 4.2.** *Let* $\mathbf{C} = \{C_1, \ldots, C_k\}$ *be an* $(r, 2r)$-*separated clustering of* $S$. *Let* $\mathsf{O}$ *be any set of size* $\varepsilon' k$. *Fixing the randomness and parameters in the algorithm with distance threshold* $r$, *let* $\widetilde{\mathbf{C}}$ *be the output on* $S$ *and* $\tilde{\mathbf{C}}'$ *be the output on* $S \cup \mathsf{O}$. *Then,* $\mathcal{A}_\gamma(\mathbf{C}, \tilde{\mathbf{C}}') \geqslant \mathcal{A}_\gamma(\mathbf{C}, \tilde{\mathbf{C}}) - \varepsilon'$.

Notice that this is optimal since $\varepsilon' k$ outliers can clearly modify $\varepsilon' k$ clusters. For DNA storage data recovery, if we desire $1 - \varepsilon$ accuracy overall, and we expect at most $\varepsilon' k$ outliers, then we simply need to aim for a clustering with accuracy at least $1 - \varepsilon + \varepsilon'$.

## 5  Experiments

We experimentally evaluate our algorithm on real and synthetic data, measuring accuracy and wall clock time. Table 1 describes our datasets. We evaluate accuracy on the real data by comparing the found clusterings to a gold standard clustering. We construct the gold standard by using the original reference strands, and we group the reads by their most likely reference using an established alignment tool (see Appendix E for full details). The synthetically generated data resembles real data distributions and properties [45]. We implement our algorithm in C++ using MPI. We run tests on Microsoft Azure virtual machines (size H16mr: 16 cores, 224 GB RAM, RDMA network).

Table 1: Datasets. Real data from Organick et. al. [45]. Synthetic data from Defn. 2.3. Appendix E has details.

| Dataset | # Reads | Avg. Length | Description |
|---|---|---|---|
| 3.1M real | 3,103,511 | 150 | Movie file stored in DNA |
| 13.2M real | 13,256,431 | 150 | Music file stored in DNA |
| 58M real | 58,292,299 | 150 | Collection of files (40MB stored in DNA; includes above) |
| 12M real | 11,973,538 | 110 | Text file stored in DNA |
| 5.3B synthetic | 5,368,709,120 | 110 | Noise $p = 4\%$; cluster size $s = 10$. |

### 5.1  Implementation and Parameter Details

For the edit distance threshold, we desire $r$ to be just larger than the cluster diameter. With $p$ noise, we expect the diameter to be at most $4pm$ with high probability. We conservatively estimate $p \approx 4\%$ for real data, and thus we set $r = 25$, since $4pm = 24$ for $p = 0.04$ and $m = 150$.

For the binary signatures, we observe that choosing larger $q$ separates clusters better, but it also increases overhead, since $\sigma_q(x) \in \{0, 1\}^{4^q}$ is very high-dimensional. To remedy this, we used a blocking approach. We partitioned $x$ into blocks of 22 characters and computed $\sigma_3$ of each block, concatenating these 64-bit strings for the final signature. On synthetic data, we found that setting $\theta_{low} = 40$ and $\theta_{high} = 60$ leads to very reduced running time while sacrificing negligible accuracy.

For the hashing, we set $w, \ell$ to encourage collisions of close pairs and discourage collisions of far pairs. Following Theorem C.1, we set $w = \lceil \log_4(m) \rceil = 4$ and $\ell = 12$, so that $w + \ell = 16 = \log_4 n$ with $n = 2^{32}$. Since our clusters are very small, we find that we can further filter far pairs by concatenating two independent hashes to define a bucket based on this 64-bit value. Moreover, since we expect very few reads to have the same hash, instead of comparing all pairs in a hash bucket, we sort the reads based on hash value and only compare adjacent elements. For communication, we use only the first 20 bits of the hash value, and we uniformly distribute clusters based on this.

Finally, we conservatively set the number of iterations to 780 total (26 communication rounds, each with 30 local iterations) because this led to 99.9% accuracy on synthetic data (even with $\gamma = 1.0$).

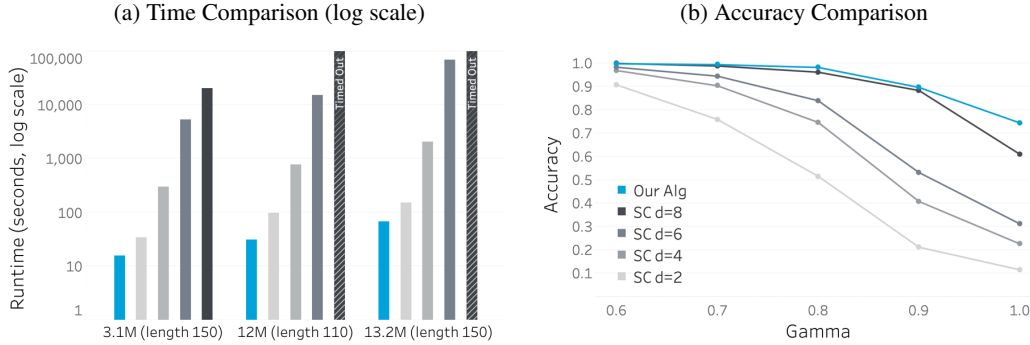

Figure 2: Comparison to Starcode. Figure 2a plots running times on three real datasets of our algorithm versus four Starcode executions using four distance thresholds $d \in \{2, 4, 6, 8\}$. For the first dataset, with 3.1M real reads, Figure 2b plots $\mathcal{A}_\gamma$ for varying $\gamma \in \{0.6, 0.7, 0.8, 0.9, 1.0\}$ of our algorithm versus Starcode. We stopped Starcode if it did not finish within 28 hours. We ran tests on one processor, 16 threads.

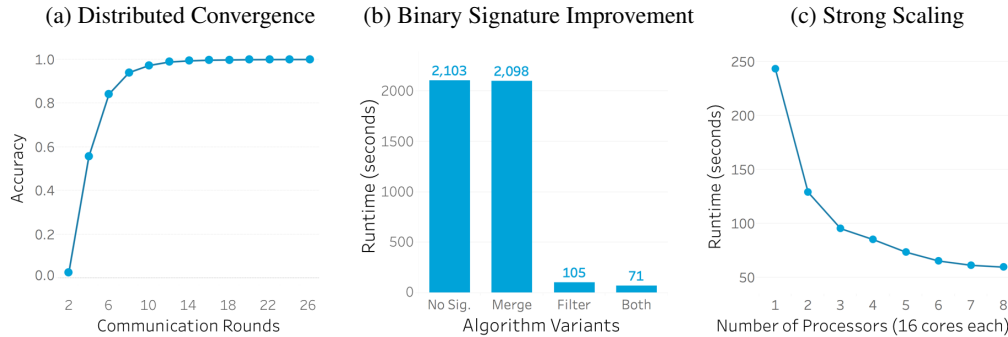

Figure 3: Empirical results for our algorithm. Figure 3a plots accuracy $\mathcal{A}_{0.9}$ of intermediate clusterings (5.3B synthetic reads, 24 processors). Figure 3b shows single-threaded running times for four variants of our algorithm, depending on whether it uses signatures for merging and/or filtering (3.1M real reads; single thread). Figure 3c plots times as the number of processors varies from 1 to 8, with 16 cores per processor (58M real reads).

**Starcode Parameters** Starcode [57] takes a distance threshold $d \in \{1, 2, \ldots, 8\}$ as an input parameter and finds all clusters with radius not exceeding this threshold. We run Starcode for various settings of $d$, with the intention of understanding how Starcode's accuracy and running time change with this parameter. We use Starcode's sphere clustering "-s" option, since this has performed most accurately on sample data, and we use the "-t" parameter to run Starcode with 16 threads.

## 5.2 Discussion

Figure 2 shows that our algorithm outperforms Starcode, the state-of-the-art clustering algorithm for DNA sequences [57], in both accuracy and time. As explained above, we have set our algorithm's parameters based on theoretical estimates. On the other hand, we vary Starcode's distance threshold parameter $d \in \{2, 4, 6, 8\}$. We demonstrate in Figures 2a and 2b that increasing this distance parameter significantly improves accuracy on real data, but also it also greatly increases Starcode's running time. Both algorithms achieve high accuracy for $\gamma = 0.6$, and the gap between the algorithms widens as $\gamma$ increases. In Figure 2a, we show that our algorithm achieves more than a 1000x speedup over the most accurate setting of Starcode on three real datasets of varying sizes and read lengths. For $d \in \{2, 4, 6\}$, our algorithm has a smaller speedup and a larger improvement in accuracy.

Figure 3a shows how our algorithm's clustering accuracy increases with the number of communication rounds, where we evaluate $\mathcal{A}_\gamma$ with $\gamma = 0.9$. Clearly, using 26 rounds is quite conservative. Nonetheless, our algorithm took only 46 minutes wall clock time to cluster 5.3B synthetic reads on 24 processors (384 cores). We remark that distributed MapReduce-based algorithms for string similarity joins have been reported to need tens of minutes for only tens of millions of reads [21, 51].

Figure 3b demonstrates the effect of binary signatures on runtime. Recall that our algorithm uses signatures in two places: merging clusters when $d_H(\sigma(x), \sigma(y)) \leqslant \theta_{low}$, and filtering pairs when $d_H(\sigma(x), \sigma(y)) > \theta_{high}$. This leads to four natural variants: (i) omitting signatures, (ii) using them for merging, (iii) using them for filtering, or (iv) both. The biggest improvement (20x speedup) comes from using signatures for filtering (comparing (i) vs. (iii)). This occurs because the cheap Hamming distance filter avoids a large number of expensive edit distance computations. Using signatures for merging provides a modest 30% improvement (comparing (iii) vs. (iv)); this gain does not appear between (i) and (ii) because of time it takes to compute the signatures. Overall, the effectiveness of signatures justifies their incorporation into an algorithm that already filters based on hashing.

Figure 3c evaluates the scalability of our algorithm on 58M real reads as the number of processors varies from 1 to 8. At first, more processors lead to almost optimal speedups. Then, the communication overhead outweighs the parallelization gain. Achieving perfect scalability requires greater understanding and control of the underlying hardware and is left as future work.

# 6 Related Work

Recent work identifies the difficulty of clustering datasets containing large numbers of small clusters. Betancourt et. al. [11] calls this "microclustering" and proposes a Bayesian non-parametric model for entity resolution datasets. Kobren et. al. [37] calls this "extreme clustering" and studies hierarchical clustering methods. DNA data storage provides a new domain for micro/extreme clustering, with interesting datasets and important consequences [12, 24, 26, 45, 52].

Large-scale, extreme datasets – with billions of elements and hundreds of millions of clusters – are an obstacle for many clustering techniques [19, 29, 33, 42]. We demonstrate that DNA datasets are well-separated, which implies that our algorithm converges quickly to a highly-accurate solution. It would be interesting to determine the minimum requirements for robustness in extreme clustering.

One challenge of clustering for DNA storage comes from the fact that reads are strings with edit errors and a four-character alphabet. Edit distance is regarded as a difficult metric, with known lower bounds in various models [1, 5, 7]. Similarity search algorithms based on MinHash [13, 14] originally aimed to find duplicate webpages or search results, which have much larger natural language alphabets. However, known MinHash optimizations [40, 41] may improve our clustering algorithm.

Chakraborty, Goldenberg, and Koucký explore the question of preserving small edit distances with a binary embedding [16]. This embedding was adapted by Zhang and Zhang [56] for approximate string similarity joins. We leave a thorough comparison to these papers as future work, along with obtaining better theoretical bounds for hashing or embeddings [17, 46] under our data distribution.

# 7 Conclusion

We highlighted a clustering task motivated by DNA data storage. We proposed a new distributed algorithm and hashing scheme for edit distance. Experimentally and theoretically, we demonstrated our algorithm's effectiveness in terms of accuracy, performance, scalability, and robustness.

We plan to release one of our real datasets. We hope our dataset and data model will lead to further research on clustering and similarity search for computational biology or other domains with strings.

For future work, our techniques may also apply to other metrics and to other applications with large numbers of small, well-separated clusters, such as entity resolution or deduplication [20, 23, 32]. Finally, our work motivates a variety of new theoretical questions, such as studying the distortion of embeddings for random strings under our generative model (we elaborate on this in Appendix B ).

# 8 Acknowledgments

We thank Yair Bartal, Phil Bernstein, Nova Fandina, Abe Friesen, Sariel Har-Peled, Christian Konig, Paris Koutris, Marina Meila, Mark Yatskar for useful discussions. We also thank Alyshia Olsen for help designing the graphs. Finally, we thank Jacob Nelson for sharing his MPI wisdom and Taylor Newill and Christian Smith from the Microsoft Azure HPC Team for help using MPI on Azure.

## Footnotes

[1]The similarity graph connects all pairs of elements with distance below a given threshold.

[2]The requirement $\gamma \in (1/2, 1]$ implies $\mathcal{A}_\gamma(\mathbf{C}, \widetilde{\mathbf{C}}) \in [0, 1]$.

[3]A cluster $C$ has diameter at most $r$ if $d_E(x, y) \leqslant r$ for all pairs $x, y \in C$.

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
