[Supplementary Material]

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

# A  Clusters are Well-Separated in Edit Distance

Let $\mathcal{U}_m$ be the uniform distribution over $\Sigma^m$. Recall that in our data model the cluster centers $c_1, \ldots, c_k$ are independently sampled from $\mathcal{U}_m$. Therefore, in this section, we analyze the edit distance between random strings. We also bound the cluster radius when clusters consist of $p$-noisy copies of the cluster centers (Definition 2.2). We will prove that a clustering $\mathbf{C} = \{C_1, \ldots, C_k\}$ is well-separated when each $C_i$ is sampled from $\mathcal{D}_{s,p,m}$ for $i \in [k]$ where we define $[k] = \{1, 2, \ldots, k\}$. The separation will hold even if the error rate is constant (that is, when $p$ is a small constant).

We need the following concentration bound for our analyses.

**Lemma A.1** ([22])**.** *Let $f$ be a function of $m$ random variables $Z_1, \ldots, Z_m$ such that $f(Z_1, \ldots, Z_m) \leqslant b$ for some b. For $i \in [m]$ let $\xi_i$ satisfy*

$$|\mathbb{E}[f \mid Z_1, \ldots, Z_{i-1}, Z_i = a_i] - \mathbb{E}[f \mid Z_1, \ldots, Z_{i-1}, Z_i = a_i']| \leqslant \xi_i.$$

*Then for any $\eta > 0$, we have that*

$$\Pr\left[|f - \mathbb{E}[f]| > \eta\right] \leqslant \exp\left(-\frac{2\eta^2}{\sum_{i=1}^{m} \xi_i^2}\right)$$

## A.1  Edit Distance of Random Strings

There exists a constant $c_{\mathsf{ind}} = c_{\mathsf{ind}}(|\Sigma|) > 0$ such that for $x, y \sim \mathcal{U}_m$,

$$\lim_{m \to \infty} \frac{d_E(x, y)}{m} = c_{\mathsf{ind}}$$

almost surely (this follows from Kingman's ergodic theorem). Determining the precise value of $c_{\mathsf{ind}}$ is a challenging open problem, related to calculating the size of a ball under edit distance [10, 25]. When $|\Sigma| = 4$, simulations suggest $c_{\mathsf{ind}} \approx 0.51$, and it is known [25] that $c_{\mathsf{ind}} > 0.338$. In what follows, we will use that $c_{\mathsf{ind}}$ is the largest constant satisfying

$$\mathbb{E}_{x,y\sim\mathcal{U}_m}\left[d_E(x, y)\right] \geqslant c_{\mathsf{ind}} \cdot m,$$

for $\Sigma = \{\mathsf{A}, \mathsf{C}, \mathsf{G}, \mathsf{T}\}$.

**Lemma A.2.** *For any $\lambda > 0$ we have*

$$\Pr_{x,y\sim\mathcal{U}_m}\left[\left|d_E(x, y) - \mathbb{E}_{x,y\sim\mathcal{U}_m}\left[d_E(x, y)\right]\right| \leqslant \lambda\sqrt{\frac{m}{2}}\right] \geqslant 1 - e^{-\lambda^2}.$$

*Proof.* Let $X = X_1 X_2 \cdots X_m$ and $Y = Y_1 Y_2 \cdots Y_m$ be random strings drawn from $\mathcal{U}_m$, and let $Z_i$ be the pair $(X_i, Y_i)$. Define $f(Z_1, \ldots, Z_m) = d_E(X, Y)$. It is easy to see that

$$|\mathbb{E}[f \mid Z_1, \ldots, Z_{i-1}, Z_i = (x_i, y_i)] - \mathbb{E}[f \mid Z_1, \ldots, Z_{i-1}, Z_i = (x_i', y_i')]| \leqslant 1,$$

where $x_i, y_i, x_i', y_i'$ are any four characters in $\Sigma$. Thus, the statement follows from Lemma A.1. $\quad\square$

The above lemma implies that random strings will have edit distance that is linear in $m$ with high probability, since their expected distance is at least $c_{\mathsf{ind}} \cdot m$ and at most $m$.

## A.2  Cluster Analysis for Edit Distance

We first analyze a cluster's radius by bounding the edit distance of $p$-noisy copies.

**Lemma A.3.** *Let $x$ be a $p$-noisy copy of $c \in \Sigma^m$. Then, for any $\lambda > 0$ we have*

$$\Pr\left[d_E(x, c) \leqslant pm + \lambda\sqrt{3m}\,\right] \geqslant 1 - e^{-\lambda^2}.$$

*Proof.* Recall that $x$ was generated from $c$ by going from left to right in $c$, and at each character, introducing an edit (substitution, insertion, or deletion) with probability $p$, independently of the other edits. Notice that the number of edits from $c$ to $x$ is binomially distributed, with parameters $m$ and $p$. Therefore, $\mathbb{E}\left[d_E(x, c)\right] = pm$, and a standard Chernoff bound proves the lemma. $\quad\square$

We tie together the above lemmas to prove that clusters are separated with high probability under edit distance when $m$ is large enough. In what follows, we think of $p$ as a small constant $p \ll c_{\text{ind}}$, to ensure separation. For high probability bounds, the number of input reads $n$ just needs to satisfy

$$n < e^{\delta' c_{\text{ind}}^2 m}$$

for a small enough constant $\delta' > 0$.

**Lemma A.4.** *Consider a clustering* $\mathbf{C} = \{C_1, \ldots, C_k\}$, *where each* $C_i \sim \mathcal{D}_{s,p,m}$, *and let* $n = sk$. *Then* $\mathbf{C}$ *is* $(r_1, r_2)$*-separated for*

$$r_1 = 2pm + 3\sqrt{m \ln n}$$
$$r_2 = c_{\text{ind}} \cdot m - 4pm - 12\sqrt{m \ln n},$$

*with probability at least* $1 - 1/n^2$.

*Proof.* Let $\mathcal{E}_1$ be the event that for all $i \in [k]$ and for all $x \in C_i$, we have

$$d_E(x, c_i) \leqslant \frac{r_1}{2},$$

where $c_i$ is the center of $C_i$. We claim that $\mathcal{E}_1$ holds with probability at least $1 - \frac{1}{2n^2}$. This follows from setting $\lambda = 2\sqrt{\ln n}$ in Lemma A.3 and a union bound over the $sk = n$ pairs $(x, c_i)$ relevant to the event $\mathcal{E}_1$. Notice that when $\mathcal{E}_1$ holds, we have that $d_E(x, x') \leqslant r_1$ for any pair $x, x' \in C_i$.

Now, recall that

$$\mathbb{E}\left[d_E(c_i, c_j)\right] \geqslant c_{\text{ind}} \cdot m,$$

and let $\mathcal{E}_2$ be the event that for all pairs of cluster centers $c_i$ and $c_j$ for $i \neq j$ we have

$$d_E(c_i, c_j) > r_2 + 2r_1.$$

When $\mathcal{E}_1 \cap \mathcal{E}_2$ holds, we have that for any two points $x \in C_i$ and $y \in C_j$ with $i \neq j$,

$$d_E(x, y) \geqslant d_E(c_i, c_j) - d_E(x, c_i) - d_E(y, c_j) > r_2 + 2r_1 - 2r_1 = r_2,$$

where the first inequality follows from the triangle inequality. By setting $\lambda = 2\sqrt{\ln n}$ in Lemma A.2 and a union bound over $\binom{k}{2} \leqslant n^2$ pairs of cluster centers, we have that $\mathcal{E}_2$ holds with probability at least $1 - \frac{1}{2n^2}$.

We conclude that $\mathcal{E}_1 \cap \mathcal{E}_2$ holds with probability at least $1 - \frac{1}{n^2}$ by a union bound, and that $\mathbf{C}$ is $(r_1, r_2)$-separated conditioned on $\mathcal{E}_1 \cap \mathcal{E}_2$. $\qquad\square$

## B  Clusters are Well-Separated Under Binary Signatures

Analogously to the previous section, we now analyze the $q$-gram distance, that is, the Hamming distance between binary signatures. We also analyze the cluster separation for random centers.

### B.1  Signature Distance of Random Strings

We start by lower bounding the expected distance for random strings.

**Lemma B.1.** *Let* $q$ *be a natural number satisfying* $4^q \geqslant 2m$ *and* $m \geqslant 2q$. *Then,*

$$\mathbb{E}_{x,y \sim \mathcal{U}_m} [d_H(\sigma_q(x), \sigma_q(y))] \geqslant 2(m - q + 1) - \frac{2(m - q + 1)^2}{4^q}.$$

*In particular, for* $q = \lceil 2 \log_4 m \rceil$ *and* $m$ *large enough,* $d_H(\sigma_q(c_i), \sigma_q(c_i)) \geqslant 2m - O(\log m)$ *in expectation for distinct cluster centers* $c_i, c_j$.

*Proof.* The number of $q$-grams in a string $x \in \Sigma^m$ is exactly $m - q + 1$. Let $X, Y$ denote the multi-set of $q$-grams in $x$ and $y$, respectively. Observe that

$$d_H(\sigma_q(x), \sigma_q(y)) = 2(m - q + 1) - 2|X \cap Y|.$$

By linearity of expectation, we simply need to prove that $\mathbb{E}[|X \cap Y|] \leqslant |X||Y|/4^q$. For each element of $X$, the probability it is contained in $Y$ is at most $|Y|/4^q$, since there are $4^q$ possible $q$-grams. Therefore the bound follows by summing over the $|X|$ $q$-grams in $X$. For distinct centers $c_i$ and $c_j$, the lower bound on the expectation of $d_H(\sigma_q(c_i), \sigma_q(c_i))$ follows by plugging in $q = \lceil 2 \log_4 m \rceil$. $\quad\square$

**Lemma B.2.** *Let $\mathcal{U}_m$ be the uniform distribution over $\Sigma^m$. Let $q$ be a natural number satisfying $4^q \geqslant 2m$ and $m \geqslant 2q$. Then, for any $\lambda > 0$, we have*

$$\Pr_{x,y \sim \mathcal{U}_m} \left[ \left| d_H(\sigma_q(x), \sigma_q(y)) - \mathbb{E}[d_H(\sigma_q(x), \sigma_q(y))] \right| \leqslant \lambda q \sqrt{m} \right] \geqslant 1 - e^{-2\lambda^2}.$$

*Proof.* Let $X = X_1 X_2 \cdots X_m$ and $Y = Y_1 Y_2 \cdots Y_m$ be random strings drawn from $\mathcal{U}_m$, and let $Z_i$ be the pair $(X_i, Y_i)$. Define $f(Z_1, \ldots, Z_m) = d_H(\sigma_q(X), \sigma_q(Y))$. Since $\sigma_q$ is defined in terms of $q$-grams, it is easy to see that

$$\left| \mathbb{E}[f \mid Z_1, \ldots, Z_{i-1}, Z_i = (x_i, y_i)] - \mathbb{E}[f \mid Z_1, \ldots, Z_{i-1}, Z_i = (x_i', y_i')] \right| \leqslant q,$$

where $x_i, y_i, x_i', y_i'$ are any four characters in $\Sigma^m$. Therefore, the statement follows from Lemma A.1. $\square$

## B.2 Cluster Analysis for Signatures

We are interested in bounding the maximum difference between the edit distance of the original strands and the Hamming distance of the signatures.

**Lemma B.3.** *Let $q$ be a natural number satisfying $0 \leqslant q \leqslant m$. For any $x, y \in \Sigma^*$,*

$$d_H(\sigma_q(x), \sigma_q(y)) \leqslant \min\{2q \cdot d_E(x, y), \ |x| + |y| - 2q + 2\}$$

*Proof.* The upper bound of $|x| + |y| - 2q + 2$ holds simply because $\sigma_q(x)$ contains at most $|x| - q + 1$ non-zero coordinates (likewise for $y$). To show $d_H(\sigma_q(x), \sigma_q(y)) \leqslant 2q \cdot d_E(x, y)$, we will analyze the case of a single edit, and prove $d_H(\sigma_q(x), \sigma_q(z)) \leqslant 2q$ when $d_E(x, z) = 1$. Then, we observe that the triangle inequality implies the upper bound for all edit distances. Let $z$ be any string in $\Sigma^*$ with $d_E(x, z) = 1$. Let $X$ and $Z$ be the set of $q$-grams in $x$ and $z$, respectively. Notice that $d_H(\sigma_q(x), \sigma_q(z))$ equals the size of the symmetric difference $|X \triangle Z|$. We claim $|X \setminus Z| \leqslant q$, since the single edit between $x$ and $z$ can cause at most $q$ elements of $X$ to be absent in $Z$. Similarly, we claim $|Z \setminus X| \leqslant q$, since the single edit between $x$ and $z$ can cause at most $q$ elements to be in $Z$ that are not present in $X$. Thus,

$$d_H(\sigma_q(x), \sigma_q(z)) = |X \triangle Z| \leqslant |X \setminus Z| + |Z \setminus X| \leqslant 2q,$$

as desired. $\square$

**Lemma B.4.** *Let $x$ be a $p$-noisy copy of $c \in \Sigma^m$. Then, for any $\lambda > 0$ we have*

$$\Pr\left[ d_H(\sigma_q(x), \sigma_q(c)) \leqslant 2q \left( pm + \lambda \sqrt{3m} \right) \right] \geqslant 1 - e^{-\lambda^2}.$$

*Proof.* This follows directly from Lemmas A.3 and B.3. $\square$

**Lemma B.5.** *Let $q = \lceil \log_4 m \rceil$. Let $\mathbf{C} = \{C_1, \ldots, C_k\}$ be a random clustering with $C_i \sim \mathcal{D}_{s,p,m}$. Then, with probability $1 - 1/n^2$, we have that*

*1. for any $i = 1, 2, \ldots, k$ and any $x, y \in C_i$,*

$$d_H(\sigma_q(x), \sigma_q(y)) \leqslant 4q \left( pm + 3\sqrt{m \ln n} \right) =: r_1'.$$

*2. for $x \in C_i$ and $y \in C_j$ with $i \neq j$,*

$$d_H(\sigma_q(x), \sigma_q(y)) \geqslant 2m - 2q \left( 4pm + 7\sqrt{m \ln n} \right) =: r_2'.$$

*Proof.* Let $\mathcal{E}_1'$ be the event that for all $i \in [k]$ and for all $x \in C_i$, we have

$$d_H(\sigma_q(x), \sigma_q(c_i)) \leqslant \frac{r_1'}{2},$$

where $c_i$ is the center of $C_i$. We claim that $\mathcal{E}_1'$ holds with probability at least $1 - \frac{1}{2n^2}$. This follows from setting $\lambda = 2\sqrt{\ln n}$ in Lemma B.4 and a union bound over the $sk = n$ pairs $(x, c_i)$ relevant

to the event $\mathcal{E}_1'$. Notice that when $\mathcal{E}_1'$ holds, we have that $d_H(\sigma_q(x), \sigma_q(x')) \leqslant r_1'$ for any pair $x, x' \in C_i$.

Now, let $\mathcal{E}_2'$ be the event that for all pairs of cluster centers $c_i$ and $c_j$ for $i \neq j$ we have

$$d_H(\sigma_q(c_i), \sigma_q(c_j)) > r_2' + 2r_1'.$$

Notice that when $\mathcal{E}_1' \cap \mathcal{E}_2'$ holds, we have that for any two points $x \in C_i$ and $y \in C_j$ with $i \neq j$,

$$d_H(\sigma_q(x), \sigma_q(y))) \geqslant d_H(\sigma_q(c_i), \sigma_q(c_j)) - d_H(\sigma_q(x), \sigma_q(c_i)) - d_H(\sigma_q(y), \sigma_q(c_j)) > r_2',$$

where the first inequality follows from the triangle inequality. By setting $\lambda = 2\sqrt{\ln n}$ in Lemma B.2 and a union bound over $\binom{k}{2} \leqslant n^2$ pairs of cluster centers, we have that $\mathcal{E}_2'$ holds with probability at least $1 - \frac{1}{2n^2}$.

We conclude that $\mathcal{E}_1' \cap \mathcal{E}_2'$ holds with probability at least $1 - \frac{1}{n^2}$ by a union bound, and that $\mathbf{C}$ satisfies the claimed bounds conditioned on $\mathcal{E}_1' \cap \mathcal{E}_2'$. $\qquad\square$

Lemma B.5 proves that the clusters are separated according to the binary signatures with high probability whenever $p \ll 1/\log m$. This is a stricter requirement than we needed for the edit distance separation, since that tolerated error rate $p = O(1)$. Constructing an efficient binary embedding for $p = \Omega(1)$ would indeed be interesting.

## B.3   Metric Embedding Related Work

We briefly mention a connection to metric embeddings for non-repetitive strings. Charikar and Krauthgamer [17] call $x \in \Sigma^m$ a *t-non-repetitive* string if all of its $t$-grams are distinct. They provide an embedding into the $\ell_1$ metric space with distortion $O(t \log m)$ for the submetric of edit distance corresponding to considering only $t$-non-repetitive stings. For random strings, we prove in Lemmas B.2 and B.3 that the binary signatures $\sigma_q(x)$ provide an embedding into Hamming space with distortion $O(\log m)$ with high probability when $q = 2\log_4 m$. It is natural to wonder if our binary signatures work in general for $O(\log m)$-non-repetitive strings (since random strings will have this property with high probability). Unfortunately, it is easy to construct pairs of example strings with linear edit distance but whose signatures have only logarithmic Hamming distance ($\approx q$). Therefore, it is an interesting open question to find an efficient metric embedding into $\ell_1$ that has distortion $O(1)$ for strings under our random model. Any such embedding must crucially use that the strings are random since Andoni and Krauthgamer [5] prove that embedding 1-non-repetitive strings into $\ell_1$ requires distortion $\Omega(\log m / \log\log m)$. Andoni and Krauthgamer also study a related question, about approximately computing edit distance under a random model [6].

## C   Theoretical Guarantees of Our Algorithm

In this section, we estimate the number of strand comparisons performed throughout the execution of the algorithm. We analyze a serial version of the algorithm that slightly differs from the one described in Section 3. This version works as follows. It maintains a collection of clusters, which we will call groups to differentiate them from the clusters in the true clustering. The algorithm starts with singleton groups i.e., initially, every strand belongs to its own group. At every iteration $t$, the algorithm picks an anchor – a random string $a(t)$ of length $L_A$. Then, in each group $g$ it picks a random strand $x_g(t)$, which we call the center of the group $g$. For every center $x$, the algorithm finds all substrings of length $L = L_A + L_H$ with prefix $a(t)$. These substrings are hash values for $g$. Denote the set of hash values by $H_g(t)$. The algorithm adds $g$ to *all buckets* of a hash table indexed by $h \in H_g(t)$. (Note that the only difference between this algorithm and the algorithm we described earlier is that in that version we add $g$ only to one of the buckets.) Then, the algorithm compares every two elements in each bucket and merges those groups whose centers are nearby with respect to the edit distance.

**Theorem C.1.** *There exists absolute constants $p_0 > 0$, $\beta_1, \beta_2, \beta_3 > 0$, such that for every $p \leqslant p_0$, $\varepsilon > 0$, integer $s \geqslant 1$, and sufficiently large $n$, $m$ ($n > m^2$) the following holds. Let $L_A = \lceil \log_4 m \rceil$, $L = \min(\lfloor \frac{\log m}{6p} \rfloor, \log_4(n/m))$. Then, after $T = \frac{\beta_1 4^{L_A} \log(s/\varepsilon)}{m(1-2p)^L}$ steps, the algorithm recovers $(1 - \varepsilon)$*

*fraction of all clusters in expectation. The expected number of comparisons performed by the algorithm is upper bounded by*

$$O\Big(\frac{n^2 m(1 + \log(s/\varepsilon)/s)}{4^L(1-2p)^L}\Big).$$

*If $L = \lfloor \frac{\log m}{6p} \rfloor$, we get the bound*

$$O\left(\frac{n^2(1 + \log(s/\varepsilon)/s)}{m^{\beta_2/p}}\right).$$

*If $L = \log_4(n/m)$, we get the bound*

$$O\Big(n^{1+2p}m^{2(1+p)}(1 + \log(s/\varepsilon)/s)\Big).$$

*Proof.* As discussed before, the true clusters $C_i$ are separated, so we never merge groups from different true clusters. Hence, every group $g$ is a subset of some cluster $C_i$. Let us introduce some notation: Denote the set of groups a cluster $C_i$ is split into at the beginning of iteration $t$ by $\mathcal{G}_i(t)$; the set of centers chosen for these groups by $\text{center}_i(t)$, and the number of these groups/centers by $s_i(t)$. Note that $\mathcal{G}_i(0)$ is a set of singletons $\mathcal{G}_i(0) = \{\{u\} : u \in C_i\}$; $\text{center}_i(0) = C_i$; and $s_i(0) = |C_i| = s$. Let $T$ be the total number of steps performed by the algorithm; and $\mathcal{G}_i(T)$, $\text{center}_i(T)$, and $s_i(T)$ be the set of groups, the set of centers, and the number of centers (respectively) in the cluster $i$ at the end of the algorithm.

We need to show that the expected number of $i$ such that $s_i(T) = 1$ is $(1-\varepsilon)n$ and then give a bound on the expected number of comparisons.

To analyze the algorithm we need to obtain lower and upper bounds on the probability that at one iteration two distinct centers $u$ and $v$ end up in the same bucket. For any two strands $u$ and $v$, let $S_{uv}$ be the set of all strings of length $L$ that are substrings of both $u$ and $v$; and let $A_{uv}$ be the set of prefixes of strings in $S_{uv}$ of length $L_A$. That is,

$$\begin{aligned} S_{uv} &= \{s \in \Sigma^L : s \text{ is a substring of } u, \text{ and } s \text{ is a substring of } v\}; \\ A_{uv} &= \{\text{prefix of length } L_A \text{ of } s : s \in S_{uv}\}. \end{aligned}$$

Suppose that at iteration $t$, strands $u$ and $v$ are chosen as centers of two distinct groups $g_1$ and $g_2$. Then, the strands $u$ and $v$ are placed into the same bucket of the hash table if and only if the anchor $a(t)$ belongs to the set $A_{uv}$. This happens with probability $|A_{uv}|/4^{L_A}$, since the anchor $a(t)$ is a random string of length $L_A$, and there are exactly $4^{L_A}$ strings of length $L_A$. Recall that in this version of the algorithm, a strand may be placed not in one but a few different buckets if the anchor string occurs in it several times. The expected number of buckets that contain both $u$ and $v$ at step $t$ is $|S_{uv}|/4^{L_A}$.

The sets $A_{uv}$ and $S_{uv}$ are random; and the quantities $|A_{uv}|$ and $|S_{uv}|$ are random variables. We estimate the expectation of $|S_{uv}|$ and $|A_{uv}|$.

**Claim C.2.** *Suppose $4^{L_A} \geqslant 6m$. Then the following bounds hold.*

1. *If $u, v \in C_i$ for some $i$, then $\mathbb{E}|S_{uv}| \geqslant (m - L + 1)(1 - 2p)^L$ and $\mathbb{E}|A_{uv}| \geqslant 1/6\,(m - L + 1)(1 - 2p)^L$.*

2. *If $u \in C_i$, $v \in C_j$ for $i \neq j$, then $\mathbb{E}|S_{uv}| \leqslant m^2 4^{-L}$.*

*Proof.* 1. If $u$ and $v$ belong to the same true cluster $C_i$, then they are noisy reads/copies of the same strand $w$. The strand $w$ has length $m$ and contains $(m - L + 1)$ substrings of length $L$. Each of these substring is present in both $u$ and $v$ with probability $(1 - p)^L \times (1 - p)^L \geqslant (1 - 2p)^L$. Hence, the expected number of common substrings is at least $(1 - 2p)^L(m - L + 1)$.

We now estimate $\mathbb{E}|A_{uv}|$. The strand $w$ contains $(m - L + 1)$ prefixes of length $L_A$ of substrings of length $L$. Let $Z_a$ be the number of occurrences of a fixed "anchor" $a$ (i.e., a substring of length $L_A$) in $w$ that starts at least $L$ characters before the end of the strand $w$. The expectation of $Z_a$ equals $\mathbb{E}[Z_a] = (m - L + 1)4^{-L_A}$, we denote this expectation by $\mu$. A standard computation shows that the

variance $\mathbf{Var}[Z_A]$ is upper bounded by $\mu + 2/3\mu$. Thus, $\mathbb{E}[Z_A^2] = \mathbf{Var}[Z_a] + (\mathbb{E}[Z_A])^2 \leqslant \mu + 2/3\mu + \mu^2$. Let $I(Z_a \geqslant 1)$ be the indicator of the event $\{Z_a \geqslant 1\}$. Then, by Cauchy–Schwarz,

$$\Pr(Z_a \geqslant 1) = \mathbb{E}[I(Z_a \geqslant 1)] = \mathbb{E}[I(Z_a \geqslant 1)^2] \geqslant 2\mathbb{E}[I(Z_a \geqslant 1)Z_a] - \mathbb{E}[Z_a^2] \geqslant$$
$$\geqslant 2\mu - (\mu + 2\mu/3 + \mu^2) = \mu/3 - \mu^2 \geqslant \mu/6.$$

Thus, the expected number of distinct anchors in strand $w$ is at least $\mu/6 \times 4^{L_A} = (m - L + 1)/6$. Each of these anchors gets copied to $u$ and $v$ without errors together with the next $L_H = L - LA$ characters in $w$ with probability at least $(1 - 2p)^L$. Hence, $\mathbb{E}|A_{uv}| \geqslant (m - L + 1)(1 - 2p)^{L_A}/6$.

2. Similarly, if $i \neq j$, then each $u$ and $v$ contains $m - L + 1 \leqslant m$ substrings of length $L$. The number of pairs of strings of length $L$ – one from $u$ and one from $v$ – is at most $m^2$. The probability that two particular substrings are the same is $4^{-L}$ (as strings $u$ and $v$ are independent random strings in the alphabet $\Sigma$). Hence the expected number of common substrings of length $L$ is at most $m^2 4^{-L}$.  $\square$

We would like to show now that the random variables $|S_{uv}|$ and $|A_{uv}|$ are concentrated around the mean w.h.p. That is true for $|A_{uv}|$ if $u, v \in C_i$ (see Lemma C.3). However, that is not true for $|S_{uv}|$ if $u \in C_i$, $v \in C_j$. To overcome this problem, we consider a sum of many random variables $|S_{uv}|$. We fix a cluster $C_i$ and vertex $u \in C_i$ and bound the sum $\sum_{v \notin C_i} |S_{uv}|$ (see Lemma C.3).

**Lemma C.3.** *There exists an absolute constant $\beta$ such that the following inequalities hold if $L < m/2$ and $L_A \geqslant \log_4 6m$.*

1. *If $u, v \in C_i$ for some $i$, then*

$$\Pr(|S_{uv}| \geqslant 1/10\, m(1 - 2p)^L) \geqslant 1 - \exp(-\beta m(1 - 4p)^L/L^2);$$

2. *If $u \in C_i$, $v \in C_j$ for some $i \neq j$, then*

$$\Pr\Big(\sum_{v \notin C_i} |S_{uv}| \leqslant \frac{2nm^2}{4^L}\Big) \geqslant 1 - m^2 \exp\Big(-\frac{\beta n}{4^L}\Big).$$

The proof of Lemma C.3 is similar to the proof of Claim C.2. We estimate the expectation as in Claim C.2 and then apply a concentration inequality. The only minor complication is that not all random variables in the sum we get are independent. Thus, in part (1), we use McDiarmid's Bounded Difference Inequality; and in part (2), we use the standard Chernoff bound, but we break the sum into $m^2$ sums of independent random variables. We omit the details here. We now proceed to the proof of Theorem C.1.

**Correctness.** We first show that the expected number of clusters completely recovered after $T$ steps of the algorithm is $(1 - \varepsilon)$. Consider a cluster $C_i$ and $u, v \in C_i$. Suppose that $u$ and $v$ are centers of two different groups in $C_i$. If the algorithm places $u$ and $v$ in the same bucket of the hash table, which happens with probability $|A_{uv}|/4^{L_A}$, then the groups corresponding $u$ and $v$ are merged, since all clusters are separated (as discussed in Sections 3 and A). We say that $C_i$ is good if for all $u, v \in C_i$ we have $|A_{uv}| \geqslant 1/10\, m(1 - 2p)^L$; otherwise, we say that $C_i$ is bad.

Lemma C.3 asserts that the expected fraction of bad clusters is at most $\exp(-\beta m(1 - 4p)^L)$. Observe that

$$(1 - 4p)^L = \exp(-L\log(1/(1 - 4p))) \geqslant \exp(-5Lp) \geqslant \exp(-5/6 \log m) = m^{-5/6}.$$

Here we used that for a sufficiently small $p$, we have $\log(1/(1 - 4p) \leqslant 5p$. We also substituted $L = \lfloor \frac{\log m}{6p} \rfloor$. Hence, the expected fraction of bad clusters is upper bounded by $\exp(-\beta m^{1/6}/L^2)$, which is much less than $\varepsilon$ for a sufficiently large $m$. Thus, we can ignore bad clusters.

Consider a good cluster $C_i$. At every iteration $t$, for any two centers $u, v \in \mathrm{center}_i(t)$, the probability that the algorithm puts $u$ and $v$ into the same bucket is $|A_{uv}|4^{-L_A}$, which at least

$$\alpha = \frac{m(1 - 2p)^L}{10 \cdot 4^{L_A}}.$$

Therefore, using Lemma C.5 from Section C.1, we obtain the following bound on the expected number of non recovered clusters:

$$s(1-\alpha)^T = s\Big(1 - \frac{m(1-2p)^L}{10 \cdot 4^{L_A}}\Big)^T \leqslant \varepsilon/2.$$

The last inequality holds, since

$$T = \frac{\beta_1 4^{L_A} \log(s/\varepsilon)}{m(1-2p)^L}.$$

**Number of Comparisons.** We now upper bound the total number of comparisons. We divide all strand comparisons into two types: internal and external. We say that a comparison of $u$ and $v$ is internal if $u$ and $u$ belong to the same true cluster $C_i$; we say that it is external, if $u$ and $v$ belong to two distinct clusters $C_i$ and $C_j$. It is easy to see that the total number of internal comparisons for a cluster $C_i$ is bounded by $|C_i| - 1 = s - 1$, since after $s - 1$ comparisons the algorithm will merge all vertices in $C_i$ into one group. Thus, the total number of internal comparisons is $k(s-1) < n$. Below, we prove Lemma C.4 that gives a bound on the number of external comparisons. This bound together with the bound on the number of internal comparisons gives us the desired result. □

**Lemma C.4.** *The expected number of external comparison performed by the algorithm in $T$ steps is upper bounded by*

$$O\Big(\frac{n^2 m(1 + \log(s/\varepsilon)/s)}{4^L(1-2p)^L}\Big).$$

*Proof.* Fix a cluster $C_i$. We estimate the expected number of external comparisons between strands in $C_i$ and outside of $C_i$. Consider a step $t$ of the algorithm. If $u$ is chosen as a center of a group in $C_i$, then the expected number of external comparisons between $u$ and strands $v$ outside of $C_i$ is upper bounded by $4^{-L_A} \sum_{v \notin C_i} |S_{uv}|$ (note: as we discussed earlier $4^{-L_A}|S_{uv}|$ is the expected number of buckets in which both $u$ and $v$ are placed at iteration $t$ if $u$ and $v$ are centers). Here, we use a conservative estimate: instead of counting only strands $v$ that are centers of groups outside of $C_i$ we count all $v$'s outside of $C_i$. Summing over all centers $u \in \text{center}_i(t)$, we get a bound on the expected number of comparisons between centers in $C_i$ and other centers:

$$4^{-L_A} \mathbb{E}\Big[\sum_{u \in \text{center}_i(t)} \sum_{v \notin C_i} |S_{uv}|\Big] \quad \leqslant \quad 4^{-L_A} \mathbb{E}\Big[|\text{center}_i(t)| \max_{u \in C_i} \sum_{v \notin C_i} |S_{uv}|\Big] \qquad (1)$$

$$= \quad 4^{-L_A} \mathbb{E}\Big[s_i(t) \sum_{v \notin C_i} |S_{uv}|\Big]. \qquad (2)$$

Let $\mathcal{E}_u$ be the event $\{\sum_{v \notin C_i} |S_{uv}| \leqslant 2nm^2/4^L\}$, and

$$\mathcal{E}_{C_i} = \cap_{u \in C_i} \mathcal{E}_u = \Big\{\max_{u \in C_i} \sum_{v \notin C_i} |S_{uv}| \leqslant \frac{2nm^2}{4^L}\Big\}.$$

Let $I_i$ be the indicator of the event $\mathcal{E}_{C_i}$. By Lemma C.3, $\Pr(\neg \mathcal{E}_u) \leqslant m^2 \exp(-\beta n/4^L)$. Using the union bound, we get $\mathbb{E}[1 - I_i] = \Pr(\neg \mathcal{E}_{C_i}) \leqslant sm^2 \exp(-\beta n/4^L) \leqslant 4^{-L}/s$. The last inequality follows from the bound $L \leqslant \log_4(n/m)$ for sufficiently large $m$. We now consider two cases: $I_i = 1$, then we bound the expected number of comparisons using (2); and $I_i = 0$, then we bound the number of comparisons by $sn$ (this is the maximum possible number of comparisons). We get the following bound on the number of comparisons between centers in $C_i$ and all other centers at iteration $t$:

$$\frac{1}{4^{L_A}} \mathbb{E}\Big[I_i \cdot s_i(t) \cdot \sum_{v \notin C_i} |S_{uv}|\Big] + \mathbb{E}\Big[(1 - I_i) \cdot sn\Big] \quad \leqslant \quad \frac{1}{4^{L_A}} \mathbb{E}\Big[I_i \cdot s_i(t) \cdot \frac{2nm^2}{4^L}\Big] + \mathbb{E}\Big[(1 - I_i) \cdot sn\Big]$$

$$\leqslant \quad \frac{2nm^2}{4^{L+L_A}} \mathbb{E}[s_i(t)] + \frac{n}{4^L}.$$

We now sum up this bound over all steps $t = 0, \ldots, T-1$ of the algorithm. By Lemma C.5, $\mathbb{E}\big[\sum_{t=0}^{T-1} |s_i(t)|\big] \leqslant T + s/\alpha$, where $\alpha = m(1-2p)^L/(10 \cdot 4^{L_A})$. Hence, the expected number of external comparisons for a cluster $C_i$ is upper bounded by

$$\frac{2nm^2}{4^{L+L_A}} \cdot \Big(T + \frac{10s \cdot 4^{L_A}}{m(1-2p)^L}\Big) + \frac{nT}{4^L} = O\Big(\frac{Tnm^2}{4^{L+L_A}} + \frac{snm}{4^L(1-2p)^L}\Big).$$

Summing up this bound over all $k$ clusters $C_i$, we get the bound:

$$O\Big(\frac{Tn^2m^2}{s4^{L+L_A}} + \frac{n^2m}{4^L(1-2p)^L}\Big) = O\Big(\frac{n^2m(1+\log(s/\varepsilon)/s)}{4^L(1-2p)^L}\Big).$$

□

## C.1 Single Cluster Dynamics

In this section, we analyze the evolution of a single cluster $C_i$ from the underlying true clustering throughout the execution of the algorithm. We obtain the desired bounds, we will only rely on the following fact: The probability that any two groups in $C_i$ are merged at step $t$ is at least $\alpha$. Note, however, that the events $\{g_1$ and $g_2$ are merged at step $t\}$ and $\{g_2$ and $g_3$ are merged at step $t\}$ are not independent.

**Lemma C.5.** *The following bounds hold.*

*1.*

$$\Pr(S_i(T) > 1) \leqslant (s-1)(1-\alpha)^T.$$

*2.*

$$\mathbb{E}\Big[\sum_{t=0}^{T-1} s_i(t)\Big] \leqslant T + \frac{s-1}{\alpha}.$$

*Proof.* 1. Pick a designated vertex $u$ in $C_i$. For every $v$ the probability that $u$ and $v$ are in two distinct groups after $t$ steps of the algorithm is at most $(1-\alpha)^t$ since at every step the probability that the group containing $u$ and group containing $v$ are merged is at least $\alpha$. Thus, the expected number of vertices that are not in the group containing $u$ is at most $(s-1)(1-\alpha)^t$. This is also a bound on the number of groups not containing $u$. Hence, $\mathbb{E}[s_i(t)] \leqslant 1 + (s-1)(1-\alpha)^t$, and, consequently, $\Pr(s_i(t) > 1) \leqslant \mathbb{E}[s_i(t) - 1] \leqslant (s-1)(1-\alpha)^t$.

2. Using the bound $\mathbb{E}[s_i(t)] \leqslant 1+(1-\alpha)^t$, we get $\sum_{t=0}^{T-1}(1+(s-1)(1-\alpha)^t) \leqslant T+(s-1)/\alpha$.   □

## D   Outlier Analysis

For this proof, we analyze a slightly simplified version of our algorithm that does not use signatures for merging. A similar result holds for the original algorithm, but it is more complicated to state (since it depends on $\mathcal{D}_{s,p,m}$), and this complexity does not add new insight. In particular, for this section, we will set $\theta_{low} = -1$. The other parameters of the algorithm can be arbitrary, except for $r$ which depends on the separation (and is mentioned in the theorem statement).

We need some definitions for the proof.

**Definition D.1.** Let $\mathbf{C} = \{C_1, \ldots, C_k\}$ be $(r, 2r)$-separated. A point $z$ *touches* a cluster $C_i$ if $d_E(z, x) \leqslant r$ for any $x \in C_i$. For a set $\mathsf{O}$ of outliers, say a cluster $C_i$ is *untouched by* $\mathsf{O}$ if no point in $\mathsf{O}$ touches it.

*Proof of Theorem 4.2.* We show that at most $\varepsilon'k$ terms in the sum in $\mathcal{A}_\gamma$ differ for $\widetilde{\mathbf{C}}$ versus $\widetilde{\mathbf{C}}'$. This follows from two simple claims:

1. If $C_i$ is untouched by $\mathsf{O}$, then the $i$th term is the same in the sums for both $\mathcal{A}_\gamma(\mathbf{C}, \widetilde{\mathbf{C}})$ and $\mathcal{A}_\gamma(\mathbf{C}, \widetilde{\mathbf{C}}')$.

2. Each outlier in $\mathsf{O}$ can touch at most one cluster in $\mathbf{C}$.

To see the first of the these claims, notice that if a cluster $C_i$ is untouched, then $d_E(x, z) > r$ for any $z \in \mathsf{O} \cup (S \setminus C_i)$. Therefore, no element of $C_i$ will ever merge with an element outside of $C_i$, and the output of the algorithm will be the same with respect to $C_i$ on both inputs $S$ and $S \cup \mathsf{O}$. The second claim follows directly from the $(r, 2r)$-separated property. Indeed, if an outlier $z$ has $d_E(x, z) \leqslant r$ for any point $x \in C_i$, then it must be that $d_E(y, z) > r$ for all $y \in C_j$ with $j \neq i$. Putting these claims together, at most $\varepsilon'k$ terms differ in the sums for $\mathcal{A}_\gamma$ and thus the accuracies differ by at most $\varepsilon'$, since each term is at most 1.   □

# E   Additional Information about Datasets and Experimental Setup

## E.1   Computing Environment and Implementation Details

We use dedicated virtual machines on Microsoft Azure within a single region and virtual network. They machines have Azure size H16mr. The specifications are as follows: Intel E5-2667 V3 3.2 GHz processors, utilizing 224GB DDR4 memory and 2TB SSD-based local storage. They have a dedicated RDMA backend network enabled by FDR InfiniBand network. The Linux operating system uses image Centos 7.1-HPC.

We implemented out algorithms using C++. We used the MPICH MPI-3 compilers and the Intel MPI 5.2 runtime library. We compiled with the -O2 flag. For MPI, we allocated a single rank per core, and the large communication steps of our algorithm are implemented with non-blocking windows, which support RDMA.

## E.2   Real Datasets Used for Evaluation

We provide details about the real datasets that we used for experimental evaluation. The real data came from a DNA data storage system presented by Organick et. al. [45]. We explain the way we have processed their sequencing data to generate our datasets. In particular, we explain the methods we used to produce a gold standard on which we evaluate our algorithm and Starcode.

Overall, the data from Organick et. al. [45] is the output of an Illumina NextSeq machine. They prepared and sequenced molecules of synthetic DNA that store encoded data. The details of wetlab preparation and amplification can be found in the paper by Organick et. al. [45].

To generate a gold standard for clustering, we must find the true mapping between references and reads. To do this, we used a standard biological alignment tool, the Burrows-Wheeler Aligner (BWA) [39]. We run BWA with 10 threads using the command

```
bwa mem -t 10 [reference file] [read file]
```

BWA first indexes the references (the expected synthesized DNA) to create a database of references. After the references are indexed, BWA compares the references to all reads in order to find the best mapping between the references and reads. Each reference maps to many reads, which will be the basis for the clusters.

BWA also outputs an alignment between reads and references in the "bam" file format. From this, we extract the reads that aligned successfully. More precisely, each line in a bam file contains several fields, one of which is the read. Another field in the bam file keeps track of the most similar reference for this read, or leaves this field empty if no references are found to align. For a read that aligns, the file also identifies the best alignment (that minimizes edit distance between the read and reference). We discard lines that do not align to any reference. Since reads of DNA often contain errors, as well as additional nucleotides before and after the expected strand, we used the alignment results to extract the portions of the reads that align to the references. Finally, from the BWA output, we know which reference each read is most similar to. Thus, we use this to create gold standard clusters.

We note that in our experiments, the clustering algorithms do not have access to the references, so the clusters generated using BWA should have much higher quality than anything produced by a clustering algorithm that does not have access to the references. Therefore, the alignments serve as a good gold standard.

The datasets were generated from a single sequencing run, which in total produced 58M reads after processing. The set of references for this run corresponds to about 40MB of data stored in DNA. Additionally, using the original file information, we are able to separate out three smaller datasets, each corresponding to a single encoded file. These represent a movie file, a text file, and a music file, respectively. The sizes of the these smaller files appear in Table 1.

We remark that in practice, any true DNA storage system has an additional challenge. The data retrieval pipeline must identify the noisy reads from the sequencer output. The complication arises because the sequencer may append relatively long strings of random characters on either end of each read that it outputs. Since systems cannot align to true references, they must use additional information, such as a known substring, to extract a high-quality subsection of the read for clustering.

Figure 4: For the 12M real reads dataset, the graph plots the accuracy $\mathcal{A}_\gamma$, for varying $\gamma$, of our algorithm versus Starcode (with different distance threshold settings for Starcode). Higher is better. Tests were run on one processor, 16 threads.

### E.2.1 Synthetic Datasets Used for Evaluation

We follow Definition 2.3. The datasets with generated using Python 3.6, using the default random library. We implemented the uniform noise generation with "random.uniform", picking insertion, deletion, or substitution, each with equal probability for $4\%$ total chance of error. We picked insertion and substitution characters uniformly at random in $\{A, C, G, T\}$.

### E.2.2 Sources of Errors

In DNA data storage systems, errors arise not only from sequencing, but also from synthesis and amplification. This leads to a higher overall error rate than sequencing alone, with less than a ten-fold difference between substitution and insertion/deletion rates. In particular, the majority of reads contain at least one insertion or deletion, and therefore the clustering algorithm must be tailored to edit distance. Finally, although Illumina error rates may be low, future technologies such as Nanopore sequencing will require clustering algorithms that are robust to much higher error rates.

## F   Accuracy Justification

We expound on our definition of accuracy and its relationship to DNA storage. Recall that the goal of clustering is to find the groups of reads that came from the same reference. Then, using these clusters, a method called trace reconstruction is used to determine the most likely reference for each group of noisy reads. The effectiveness of trace reconstruction depends on the number of *traces*. Therefore, larger clusters are clearly better. When the size of the original clusters varies, it is better to have more traces from small clusters, therefore we use this parameter $\gamma$ to measure the fraction of recovered strings in a cluster.

Trace reconstruction methods do not have a built-in way to handle false positives. Fortunately, it is easy to check false positives because we know the threshold $r$. In particular, we assume that any clustering methods will check false positives before outputting the clusters.

## G   Additional Experimental Result

We include an accuracy comparison in Figure 4 for the 12M real reads dataset. This corresponds to the times reported for this dataset in Figure 2a in Section 5. We show the output of Starcode for distance parameters 1 through 7 (the setting of distance 8 did not finish in 28 hours).