[Reviews · NeurIPS 2017]

Reviewer 1



AIM In this paper, the authors present a string clustering problem motivated by DNA data storage. They introduce a novel distributed clustering algorithm, with a new hashing scheme for edit distance. They present results on simulated and real data in terms of accuracy, performance, scalability, and robustness. Comments The paper is well written. It is clear and it reads well. The authors clearly state the starting point of the problem and elucidate the characteristics of the clustering problem. Moreover, they consider strategies to reduce computational time such as hashing based on the edit distance/q-gram distance. The authors should clarify how they evaluate the performance in terms of the benchmark used when not using the synthetic dataset.

Reviewer 2



The background is to store data in DNA sequences, where one can encode information into specially designed DNA sequences. The data is then retrieved using DNA sequencing machine. Since there can be sequencing errors in the retrieving process, the DNA sequences need to be specially designed so that information can be robustly retrieved. Such datasets have a lot of small clusters that grows linearly with size of information to be coded. Each cluster contain only a few sequences. The clusters differs a lot to each other, which is determined in the design process. More importantly, not all clusters need to be correctly identified in order to retrieve the original information, which mean the data storage system is very robust. The dataset is easy in the sense that the clusters are well separated. It is challenging in the sense that the size is in the magnitude of billions. The ms proposed a clustering method based on a Hashing technique described on page 4, section 3.2. The reads are then grouped locally and globally by their Hashing values, repeated for a predefined iterations. The ms provides justifications about how to construct the Hash function and the experiment result shows the method does its job. The ms also use q-gram distance to approximate the edit distance to avoid some unnecessary computations. In general I like this paper and propose to accept it. Since I am not familiar with related literature and I wander if it is possible to split the dataset in small chunks and run fast kNN (e.g. http://www.cs.ubc.ca/research/flann/)?

Reviewer 3



The paper presents a solution to a new type of clustering problem that has emerged from studies of DNA-based storage. Information is encoded within DNA sequences and retrieved using short-read sequencing technology. The short-read sequencer will create multiple short overlapping sequence reads and these have to be clustered to establish whether they are from the same place in the original sequence. The characteristics of the clustering problem is that the clusters are pretty tight in terms of edit distance (25 max diameter here - that seems quite broad given current sequencing error rates) but well separated from each other (much larger distance between them than diameter). I thought this was an interesting and timely application. DNA storage is a hot topic and this kind of clustering is one of the computational bottlenecks to applying the method. I guess the methods presented will also be of general interest to the NIPS audience as an example of a massive data inference problem. The methods look appropriate for the task and the clustering speed achieved looks very impressive. The paper is well written and the problem, methods and results were easy to follow. My main concern with the work centred around how realistic is the generative model of the data. The substitution error rate and the insertion/deletion error rates are set to be the same parameter p/3. I think that current Illumina high-throughput sequencing is associated with much higher substitution error rates than insertion/deletion rates, I think perhaps 100-fold less insertion/deletion than substitution for some illumina technologies. If insertion/deletion rates are much lower than substitution errors then perhaps the problem is less challenging than stated here since there would be less requirement for dealing with edit distances in the case insertions/deletions are rare. Also, for most technologies the error rates from sequencing are can be well established and therefore methods can be fine-tuned given knowledge of these parameters.